# Prognostic Impact of Pathologic Features in Molecular Subgroups of Endometrial Carcinoma

**DOI:** 10.3390/jpm13050723

**Published:** 2023-04-25

**Authors:** Martina Ruscelli, Thais Maloberti, Angelo Gianluca Corradini, Francesca Rosini, Giulia Querzoli, Marco Grillini, Annalisa Altimari, Elisa Gruppioni, Viviana Sanza, Alessia Costantino, Riccardo Ciudino, Matteo Errani, Alessia Papapietro, Sara Coluccelli, Daniela Turchetti, Martina Ferioli, Susanna Giunchi, Giulia Dondi, Marco Tesei, Gloria Ravegnini, Francesca Abbati, Daniela Rubino, Claudio Zamagni, Emanuela D’Angelo, Pierandrea De Iaco, Donatella Santini, Claudio Ceccarelli, Anna Myriam Perrone, Giovanni Tallini, Dario de Biase, Antonio De Leo

**Affiliations:** 1School of Anatomic Pathology, Department of Biomedical and Neuromotor Sciences, University of Bologna, 40126 Bologna, Italy; 2Department of Medical and Surgical Sciences (DIMEC), University of Bologna, 40138 Bologna, Italy; 3Solid Tumor Molecular Pathology Laboratory, IRCCS Azienda Ospedaliero-Universitaria di Bologna, 40138 Bologna, Italy; 4Pathology Unit, IRCCS Azienda Ospedaliero-Universitaria di Bologna, 40138 Bologna, Italy; 5Unit of Medical Genetics, IRCCS Azienda Ospedaliero-Universitaria di Bologna, 40138 Bologna, Italy; 6Radiation Oncology, IRCCS Azienda Ospedaliero-Universitaria di Bologna, 40138 Bologna, Italy; 7Division of Gynecologic Oncology, IRCCS Azienda Ospedaliero-Universitaria di Bologna, 40138 Bologna, Italy; 8Department of Pharmacy and Biotechnology (FaBit), University of Bologna, 40126 Bologna, Italy; 9IRCCS Azienda Ospedaliero-Universitaria di Bologna, 40138 Bologna, Italy; 10Department of Medical, Oral and Biotechnological Sciences, University “G. d’Annunzio”, 66100 Chieti-Pescara, Italy

**Keywords:** endometrial carcinoma, histopathologic parameters, prognosis, molecular classification, histopathology

## Abstract

The molecular characterization of endometrial carcinoma (EC) has recently been included in the ESGO/ESTRO/ESP guidelines. The study aims to evaluate the impact of integrated molecular and pathologic risk stratification in the clinical practice and the relevance of pathologic parameters in predicting prognosis in each EC molecular subgroup. ECs were classified using immunohistochemistry and next-generation sequencing into the four molecular classes: *POLE* mutant (*POLE*), mismatch repair deficient (MMRd), p53 mutant (p53abn), and no specific molecular profile (NSMP). According to the WHO algorithm, 219 ECs were subdivided into the following molecular subgroups: 7.8% *POLE*, 31% MMRd, 21% p53abn, 40.2% NSMP. Molecular classes as well as ESGO/ESTRO/ESP 2020 risk groups were statistically correlated with disease-free survival. Considering the impact of histopathologic features in each molecular class, stage was found to be the strongest prognostic factor in MMRd ECs, whereas in the p53abn subgroup, only lymph node status was associated with recurrent disease. Interestingly, in the NSMP tumor, several histopathologic features were correlated with recurrence: histotype, grade, stage, tumor necrosis, and substantial lymphovascular space invasion. Considering early-stage NSMP ECs, substantial lymphovascular space invasion was the only independent prognostic factor. Our study supports the prognostic importance of EC molecular classification and demonstrated the essential role of histopathologic assessment in patients’ management.

## 1. Introduction

Endometrial carcinoma (EC) represents the most common gynecological cancer in Western countries with an increased incidence in recent years [1]. Usually, surgery is the primary treatment for women with endometrial carcinoma since they present early-stage and low-risk disease [2]. However, in about 15% of cases, patients have a high-risk disease and adjuvant treatment is required [3]. Conventionally, prognostic assessment and treatment approaches have been based on the evaluation of clinicopathologic parameters (e.g., histotype, grade, stage). Among pathologic factors, lymphovascular space invasion (LVSI), recorded in approximately 35% of EC cases, is one of the strongest indicators of lymph node involvement, pelvic recurrence, and distant metastasis [4,5,6,7,8].

Recently, new insights into the molecular landscape of endometrial carcinoma are revolutionizing the diagnostic–therapeutic approach to this cancer. The integrated genomic characterization of The Cancer Genome Atlas (TCGA) has defined four distinct prognostic subgroups of endometrial carcinoma: *POLE* mutant (*POLE*) tumors with an excellent prognosis, mismatch repair deficient (MMRd) and no specific molecular profile, (NSMP) tumors with intermediate prognosis, and p53 mutant (p53abn) with poor prognosis [9]. Using methods widely available in clinical practice, these four subgroups can be determined by their surrogate markers: *POLE* sequencing and immunohistochemistry for p53 and MMR proteins, resulting in a practical and clinically useful molecular classification tool [10,11,12]. In order to enhance and personalize patient care, the European Society of Gynecological Oncology (ESGO), the European Society for Radiotherapy and Oncology (ESTRO), and the European Society of Pathology (ESP) published updated guidelines for risk group assessment in endometrial cancer in 2020. These guidelines integrate molecular markers and clinicopathologic parameters [3].

Therefore, the present study aims to evaluate the relevance of molecular classification and traditional pathological factors in a cohort of patients with endometrial carcinoma treated at a referral center. Specifically, the objectives of this study were: (1) to evaluate the role of an integrated risk assessment algorithm in clinical practice that includes both pathologic features and molecular classes; (2) to explore the prognostic impact of pathologic parameters in each EC molecular subgroup to achieve more personalized management of patients.

## 2. Materials and Methods

### 2.1. Clinicopathologic Data of EC Patients

The present study recruited all cases of primary endometrial carcinoma from January 2018 to September 2022. The patient cohort consisted of a preliminary group of retrospectively studied patients [13] and a subsequent consecutive group of prospectively enrolled patients. Cases were characterized at the time of histologic diagnosis, and follow-up data were prospectively collected. The study cohort included 219 patients that underwent surgery at the Division of Gynecologic Oncology, “IRCCS Azienda Ospedaliero-Universitaria di Bologna” (Bologna, Italy). The present study was approved by the local ethics committee CE-AVEC (Comitato Etico-Area Vasta Emilia Centro;registration n. 27/2019/Sper/AOUBo). All patients provided their written agreement to use their tissues and data for the study. According to ESGO/ESTRO/ESP guidelines, surgical management included total hysterectomy with bilateral adnexectomy surgery. Specifically, minimally invasive surgery was the preferred surgical approach for stage I/II endometrial carcinomas, while for advanced cases, abdominal hysterectomy was performed [3]. Only surgical resected cases were collected for the present study. Histology slides and all histopathologic features were carefully reviewed and assessed by two experienced pathologists (D.S., A.D.L.). For each case, a representative formalin-fixed and paraffin-embedded (FFPE) tissue block was retrieved from the archives of the Anatomic Pathology Unit of “IRCCS Azienda Ospedaliero-Universitaria di Bologna” (Bologna, Italy) and was used for immunohistochemical and molecular analyses. Complete clinicopathologic findings obtained from clinical, surgical, and pathologic records were recorded in a comprehensive database. Parameters included age at diagnosis, International Federation of Gynecology and Obstetrics (FIGO) stage, body mass index (BMI), and follow-up information.

Histologic typing was performed according to the 2020 World Health Organization (WHO) classification of female genital tumors [14]. Tumor grading was assessed by applying a 2-tiered grading system as low-grade (FIGO grade 1 and 2) and high-grade (FIGO grade 3) [15].

Lymphovascular space invasion (LVSI) was defined by the presence of tumor cells within endothelial-lined vascular/lymphatic spaces beyond the tumor invasive front. A 3-tiered system, based on the recommendations by Bosse et al. [16], was used for grading LVSI: (1) Absent: no LVSI; (2) Focal: a single focus, consisting of 1–2 vessels involved by neoplasm, identified around the tumor; (3) Substantial: diffuse or multifocal LVSI (3 or more involved vessels) recognized around the tumor, regardless of the degree of myometrial invasion without immunohistochemical staining.

The pattern of myometrial invasion was reported, specifying the presence of so-called microcystic, elongated, and fragmented (MELF) glands [17] and/or as single invasive cells or small groups of cells (tumor budding) [18].

Extensive tumor necrosis has been reported; necrosis only within the glands or on the tumor surface has not been evaluated.

The number of mitoses per 10 high-power fields (×400 magnification) served as the mitotic index’s unit of measurement.

### 2.2. Immunohistochemistry

Four-micron serial sections were cut from each paraffin block and rapidly processed in an automated Benchmark Ultra immunostainer (Ventana Medical Systems, Tucson, AZ, USA) for immunohistochemical expression of p53, PTEN, MLH1, PMS2, MSH2, and MSH6 using Ventana antibodies and OptiView DAB detection kit (brown color) (Ventana Diagnostic Systems, Tucson, AZ, USA). Sections were counterstained using Hematoxylin and Bluing reagent following Ventana indications. Immunohistochemical staining of p53 was evaluated as abnormal/mutant-like (p53abn) in case of: (i) protein overexpression, (ii) “null” phenotype, or (iii) positive cytoplasmic staining; otherwise, it was considered normal (wild-type) [19].

PTEN expression was defined: (i) positive (uniform or heterogeneous staining in the neoplastic cells); (ii) negative (no cytoplasmic/nuclear immunostaining) [20]. Mismatch repair proteins (MLH1, PMS2, MSH2, and MSH6) were scored negative if no nuclear immunostaining was present. Absence of one of the four proteins or negative staining for MLH1/PMS2 or MSH2/MSH6 defined a deficiency in mismatch repair (MMRd) [21].

### 2.3. DNA Extraction and Next-Generation Sequencing

From two to four 10-µm-thick FFPE tissue sections were used for DNA extraction, according to the neoplastic areas selected on the hematoxylin and eosin (H&E) control-stained slide. A lab-developed panel including the following genomic areas was used to amplify about 30 ng of gDNA (human reference sequence hg19/GRCh37, total of 169 amplicons, 12.74 kb): *BRAF* (exon 15), *cKIT* (exons 8, 9, 11, 13, 17), *CTNNB1* (exons 3, 7, 8), *HRAS* (exons 2–4), *KRAS* (exons 2–4), *NRAS* (exons 2–4), *PIK3CA* (exons 10, 21), *POLE* (exons 9–14), and *TP53* (exons 4–9) are among the genes with the biggest CDS region [22,23]. The amplicon libraries were sequenced with a Gene Studio S5 Prime sequencer (Thermo Fisher Scientific, Waltham, MA, USA) as previously described [22,23]. Only nucleotide variations detected in at least 5% of the total number of reads analyzed, and observed in both strands, were considered for the mutational call. Ion Reporter Software (version 5.18, ThermoFisher Scientific) and Integrative Genomics Viewer 2.12.2 (IGV) tool (Available online: http://software.broadinstitute.org/software/igv/ accessed on 1 February 2023) were used to analyze the obtained sequences. The Varsome database (https://varsome.com/, accessed on 1 February 2023), was used to evaluate the pathogenicity of each mutation.

### 2.4. Molecular Classification

*POLE*, MMRd, p53abn, and NSMP subgroups were classified according to the molecular classification included in the 5th edition of the WHO Classification of Female Genital Tumors [14]. As shown in Figure 1, all tumors were tested for *POLE* mutations to identify the *POLE* subgroup [24]. In the absence of *POLE* mutations, IHC analysis for MMR proteins was used to define MMR deficient (MMRd) tumors. Subsequently, IHC for p53 was performed to detect p53abn carcinomas. Tumors with normal IHC expression of p53 and MMR and no *POLE* mutations were defined as “no specific molecular profile” (NSMP).

### 2.5. Statistics

Summary statistics are reported as numbers (percentages) or mean ± standard deviation (SD). Chi-squared test, Fisher’s exact test, *t*-test, Mann–Whitney test, and Kruskal–Wallis test were used for comparison between groups. The Kaplan–Meier method with the log-rank test was used to create survival curves; all recurrences (local, regional, and distant) were treated as events. All given *p* values were based on two-sided tests, and statistical significance was defined as *p* < 0.05. Analyses were performed using Stata software (version 15, Stata Statistical Software: Release 15, 2017; StataCorp LLP, College Station, TX, USA).

## 3. Results

### 3.1. Characteristics of the Endometrial Carcinoma Cohort

Clinicopathologic features and molecular classification of the 219 cases of endometrial carcinoma are detailed in Table 1. According to the 2020 WHO classification of Female Genital Tumors [14], 167 (76.3%) were endometrioid carcinomas, 25 (11.4%) dedifferentiated/undifferentiated, 20 (9.1%) serous, 3 (1.4%) clear cell carcinomas, and 4 (1.8%) carcinosarcomas. Applying a 2-tiered grading system [15], 127 (58%) were low-grade tumors and 92 (42%) were high-grade tumors. Lymph node metastases were detected in 33 (15.1%) cases. The majority of cases (76.7%) were early-stage (FIGO stage I-II) carcinomas, while 42 (19.2%) cases were stage III, and 9 (4.1%) cases were stage IV at diagnosis. Median follow-up was 32.8 months (range 1–144) and disease recurrence occurred in 38 (17.4%) patients.

FIGO stage was significantly associated with disease-free survival (log-rank: χ^2^ = 54.88, *p*-value < 0.001; see Figure 2). Overall survival was not considered for the relatively short follow-up and few events.

### 3.2. Characteristics of Molecular Subgroups

Following the 2020 WHO algorithm, EC cases were classified into the following molecular subgroups: 17 (7.8%) *POLE* group, 68 (31.0%) MMRd group, 46 (21.0%) p53abn group, 88 (40.2%) NSMP group. Molecular groups are associated with different clinicopathologic parameters (BMI, histotype, grade, FIGO stage, LVSI, MELF, tumor budding, and mitoses) shown in Table 2. Sixteen cases (7.3%) were characterized by more than one molecular feature (so-called “multiple classifier” tumors): 9 were MMRd and p53 abnormal, 4 were *POLE*-mutated and p53 abnormal, 2 were triple positive (*POLE*-mutated, MMRd, and p53 abnormal), and one was *POLE*-mutated and MMRd.

Illustrative histopathologic features of EC molecular subgroups are shown in Figure 3.

Molecular subgroups were significantly correlated with disease-free survival (log-rank: χ^2^ = 26.07, *p*-value < 0.001; see Figure 4). Specifically, no *POLE*-mutated cases relapsed, while disease recurrence was identified in 19 (41.3%) p53abn, 10 (11.4%) NSMP, and 9 (13.2%) MMRd cases.

According to ESGO/ESTRO/ESP 2020 guidelines integrated with molecular subgroups, cases were subdivided into five prognostic risk groups: 91 (41.5%) low-risk, 19 (8.7%) intermediate-risk, 23 (10.5%) high-intermediate, 77 (35.2%) high-risk and 9 (4.1%) advanced/metastatic. ESGO/ESTRO/ESP 2020 risk groups were significantly correlated with disease-free survival (log-rank: χ^2^ = 57.51, *p*-value < 0.001; see Figure 5).

### 3.3. Histopathologic Parameters in Molecular Subgroups

After defining the prognostic role of molecular classification and ESGO/ESTRO/ESP risk groups, we evaluated the impact of histopathologic features in each molecular subgroup in order to assess the association with disease recurrence. These correlations were not conducted for the *POLE* subgroup because there have not been any disease recurrences.

In MMRd subgroup, FIGO stage and lymph node status were correlated with disease recurrence (log-rank: χ^2^ = 27.95, *p*-value < 0.001; see Figure 6, and log-rank: χ^2^ = 4.33, *p*-value = 0.033).

In p53abn subgroup, only lymph node status was associated with disease recurrence (log-rank: χ^2^ = 4.76, *p*-value = 0.029; see Figure 7).

In NSMP, several histopathologic parameters were associated with disease recurrence: histotype (log-rank: χ^2^ = 12, *p*-value = 0.001; see Figure 8A), FIGO stage (log-rank: χ^2^ = 29.77, *p*-value < 0.001; see Figure 8B), grade (log-rank: χ^2^ = 21.78, *p*-value < 0.001; see Figure 8C), lymph nodes status (log-rank: χ^2^ = 22.45, *p*-value < 0.001; see Figure 8D), mitoses (log-rank: χ^2^ = 8.07, *p*-value = 0.002), extensive tumor necrosis (log-rank: χ^2^ = 15.99, *p*-value < 0.001; see Figure 8E), LVSI (log-rank: χ^2^ = 24.03, *p*-value = 0.033; see Figure 8F).

Considering NSMP subgroup, on multivariable analysis, dedifferentiated histotype, FIGO stage, and extensive tumor necrosis were independently associated with disease recurrence (see Table 3).

Considering only early-stage (FIGO stage I and II) NSMP carcinomas, multivariate analysis showed lymphovascular space invasion results to be an independent prognostic factor (see Table 4).

## 4. Discussion

The molecular classification of endometrial carcinoma emerged from The Cancer Genome Atlas (TCGA) study provides attractive and useful prognostic insights with relevant diagnostic and therapeutic impact. The molecular subgrouping is considered potentially superior to conventional typing and histologic grading. However, this classifier does not replace clinicopathologic risk assessment based on conventional histopathologic parameters. In this regard, the recent ESGO/ESTRO/ESP guidelines have proposed risk assessment based on the integration of clinicopathologic and molecular features [3,25]. Furthermore, some studies have proposed several molecular alterations, not included in current risk stratification, that have been found to be associated with the clinical outcomes of EC, such as *CTNNB1* exon 3 mutations, L1CAM overexpression, lack of estrogen receptor (ER) and progesterone receptor (PR) expression, ARID1A alterations, chromosome 1q amplification or other copy number alterations, miRNA expression [13,23,26,27,28,29,30,31]. However, the prognostic relevance of these molecular alterations in the context of the EC molecular classification is not well clarified. In the present work, we analyzed a cohort of patients with primary endometrial carcinoma and treated at a tertiary referral center demonstrating how molecular subgroups of endometrial carcinoma and histopathologic parameters work better together. Expanding our group’s previously published data, we prospectively increased the study cohort by investigating the prognostic role of molecular classification and histopathologic characteristics [25]. Specifically, we have consolidated the reported features of molecular subgroups of endometrial carcinoma. *POLE* mutated carcinomas are characterized by lower age at diagnosis, low BMI, and are usually high-grade, endometrioid, and undifferentiated/dedifferentiated histotype. MMRd subgroup has similar features to *POLE* carcinomas, but patients have older age at diagnosis and higher BMI. MMRd tumors are associated with substantial LVSI and a higher rate of lymph node metastases. p53abn carcinomas arise in older patients with low BMI, tumors are high-grade, histologically serous, endometrioid, and carcinosarcoma, and are characterized by aggressive histopathologic parameters (diffuse LVSI, lymph node metastasis, advanced stage). NSMP tumors represent the majority of cases, related to higher BMI, and they are predominantly early-stage low-grade endometrioid carcinomas. Consistently with literature data, we also demonstrated that molecular subgroups have a significant prognostic impact in our cohort: *POLE* carcinomas had excellent prognosis without any recurrent event, MMRd and NSMP carcinomas showed intermediate prognosis, and p53abn carcinomas presented worse prognosis. The heterogeneity of the NSMP group comprising both relatively indolent and aggressive ECs also emerges in our cohort. Furthermore, according to the ESGO/ESTRO/ESP guidelines, the combination of clinicopathologic and molecular characteristics has been proven to be essential in accurately assessing prognostic risk classes [25]. In addition, the present study aimed to evaluate the association of histopathologic parameters with disease recurrence in each molecular group. This objective emerged from trying to clarify whether conventional histopathologic features could have clinical relevance to date in the setting of EC molecular classification. Concerning MMRd and p53abn subgroups, FIGO stage and therefore lymph node involvement were the only factors associated with disease recurrence. Interestingly, in NSMP class, which represents the most various tumor group also in our study, different histopathologic parameters have a significant prognostic impact. Stage, histologic typing, and the presence of extensive tumor necrosis were found to be associated with disease recurrence, having instead LVSI marginal significance. Restricting the analysis to early-stage NSMP carcinomas, LVSI was the only independent prognostic factor. Our results support the vital and essential value of accurate assessment of pathologic parameters for better prognostic refinement, especially in the NSMP subgroup. This study reveals that despite the attractive and elegant proposal of additional molecular prognostic markers, an accurate histopathologic assessment will remain a cornerstone, cost-effective and applicable worldwide, and will continue to be critical for patient management. The lack of agreement in the assessment of these parameters is well documented in the literature [32,33,34,35] and to overcome these limitations, the International Society of Gynecological Pathologists has developed practical and applicable recommendations focused on the proper evaluation of histopathologic features to minimize disagreement and promote uniformity in the approach to their recognition [36]. Among different histologic parameters, recent studies have emphasized that lymphovascular space invasion (LVSI) is an independent prognostic factor that influences the risk of recurrence and, therefore, the indications for adjuvant treatment [16,37,38]. Other recent studies have shown that grading the extent of LVSI further improves its prognostic value in patients with stage I endometrioid EC [39,40]. The results of our study are consistent with this evidence and showed that the strength of LVSI is different depending on the molecular subgroup. Specifically, we found that substantial LVSI is associated with significantly higher rates of disease recurrence in early-stage NSMP tumors; in contrast, focal LVSI was not correlated to an increased risk of recurrence, confirming that the extent of LVSI plays an important role in determining the outcome. LVSI was not relevant in *POLE* carcinomas and did not reach statistical significance in MMRd and p53abn subgroups. Our novel finding that substantial LVSI is an independent adverse prognostic factor in stage I/II NSMP carcinomas implies that this parameter would be reproducible among pathologists, as demonstrated by recent work [41], and its role would be incorporated into decision-making algorithms for adjuvant treatment. Furthermore, as previously demonstrated, conventional pathologic features (histotype, grade, myometrial invasion, and LVSI) were proven to be significant prognostic parameters to correctly assess ESGO/ESTRO/ESP risk classes in the majority of cases (~60%) regardless of molecular classification [25]. However, the limitations of our study are the few recurrent disease events and the relatively short follow-up. A longer follow-up and a larger cohort might allow further consideration of histopathologic parameters in the other molecular subgroups and a better definition of the impact of these factors on overall survival.

## 5. Conclusions

In conclusion, our study showed the importance of applying molecular classification of endometrial carcinoma for precision medicine; however, to date, the accurate and consistent reporting of pathologic parameters, including the evaluation of LVSI, is still vital to ensure optimal patient management.

## Figures and Tables

**Figure 1 jpm-13-00723-f001:**
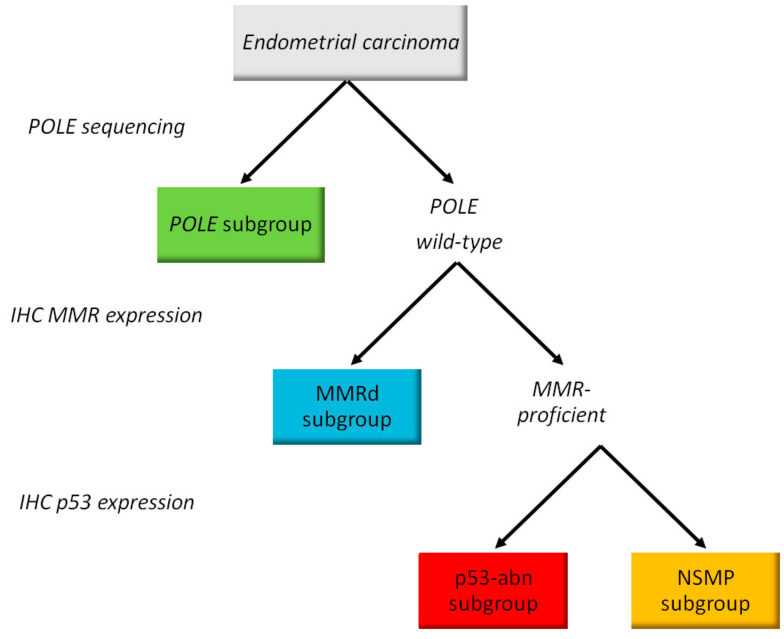
Diagnostic algorithm for surrogate molecular classification of endometrial carcinoma classification (all histotypes including carcinosarcoma).

**Figure 2 jpm-13-00723-f002:**
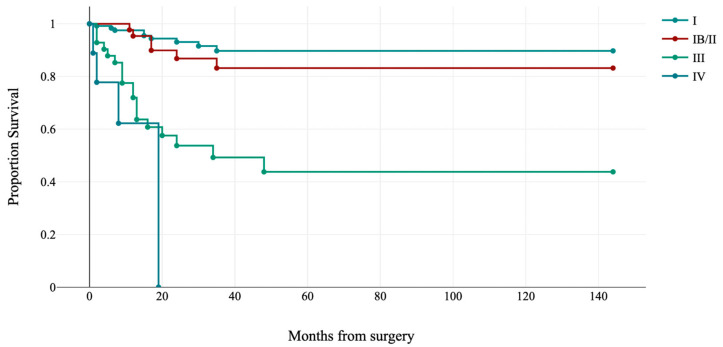
Kaplan–Meier estimates of disease-free survival by FIGO stage (*p* < 0.001).

**Figure 3 jpm-13-00723-f003:**
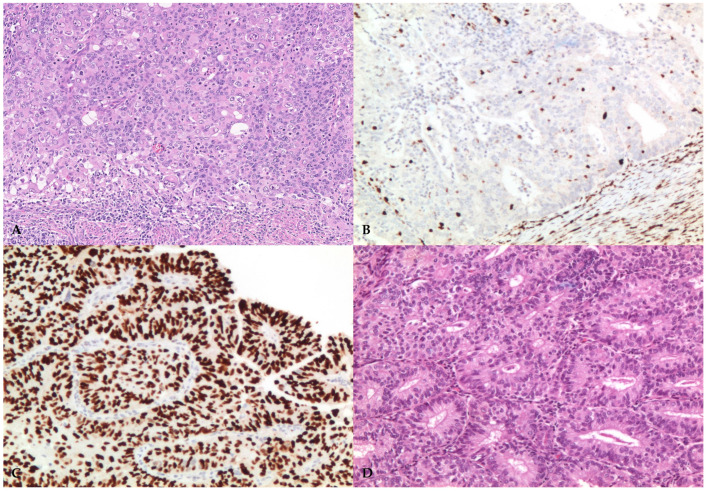
*POLE*-mutated carcinoma (**A**); MMRd carcinoma (**B**); p53abn carcinoma (**C**); NSMP carcinoma (**D**); ((**A**) ×100 magnification, (**B**–**D**) ×200 magnification).

**Figure 4 jpm-13-00723-f004:**
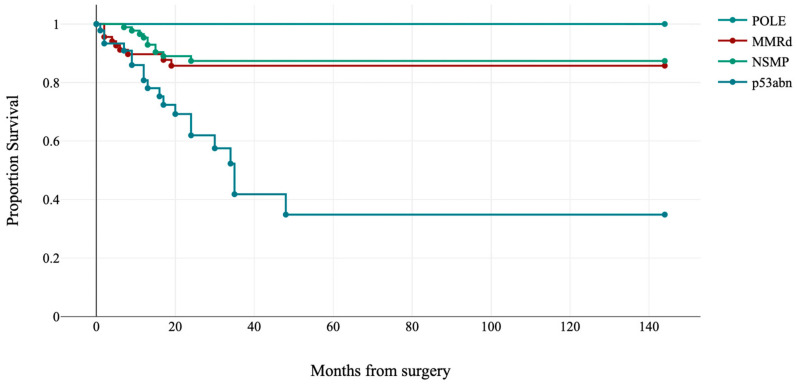
Kaplan–Meier estimates of disease-free survival by molecular subgroups (*p* < 0.001).

**Figure 5 jpm-13-00723-f005:**
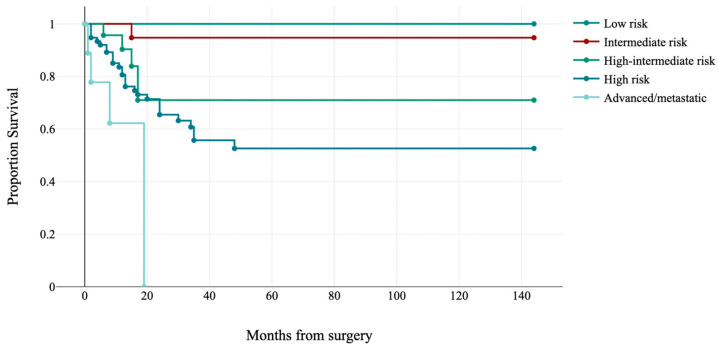
Kaplan–Meier estimates of disease-free survival by ESGO/ESTRO/ESP risk groups (*p* < 0.001).

**Figure 6 jpm-13-00723-f006:**
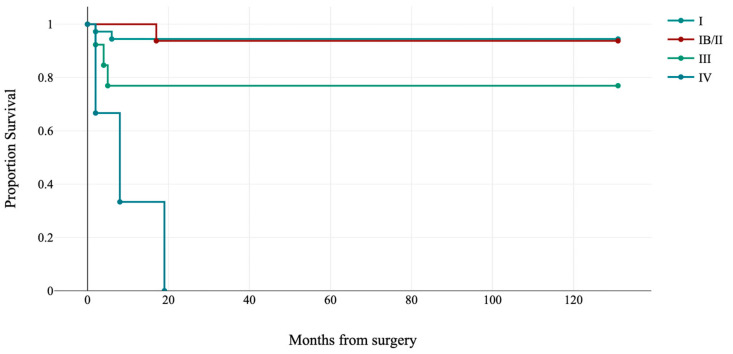
Kaplan–Meier estimates of disease-free survival by FIGO stage in MMRd subgroup (*p* < 0.001).

**Figure 7 jpm-13-00723-f007:**
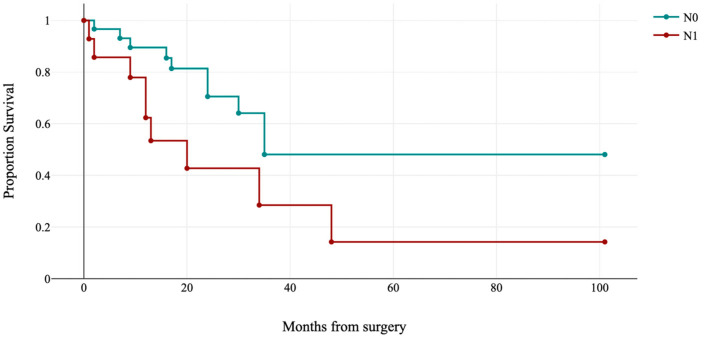
Kaplan–Meier estimates of disease-free survival by lymph node status (N0: lymph nodes negative; N1: lymph nodes positive) in the p53abn subgroup (*p* = 0.029).

**Figure 8 jpm-13-00723-f008:**
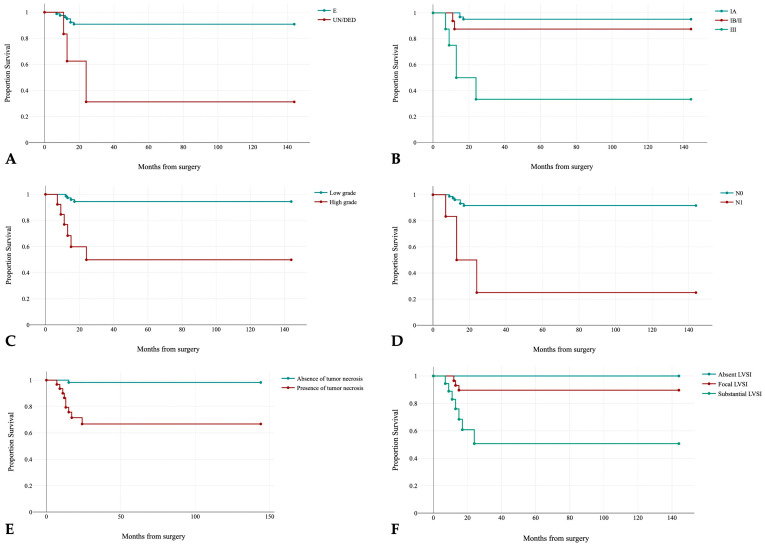
Kaplan–Meier estimates of disease-free survival in the NSMP subgroup by histotype (E: endometrioid, UN/DED: undifferentiated/dedifferantiated) *p* < 0.001 (**A**); FIGO stage *p* < 0.001 (**B**), grade *p* < 0.001 (**C**), lymph nodes status (N0: absence of lymph nodes metastasis, N1: presence of lymph nodes metastasis) *p* < 0.001 (**D**), extensive tumor necrosis *p* < 0.001 (**E**); LVSI *p* = 0.033 (**F**).

**Table 1 jpm-13-00723-t001:** Clinicopathologic characteristics and molecular subgroups of the cohort. Values are counts (percentages) or mean ± standard deviation (interquartile range).

Characteristics of EC Cases		*n* = 219 (%)
Age, years		62.5 ± 10.4
		(34–86)
Body mass index, kg/m^2^		28.11 ± 7.2
		(18.2–55.3)
Histotype		
Endometrioid		167 (76.3)
Dedifferentiated/ Undifferentiated		25 (11.4)
Serous		20 (9.1)
Clear cell		3 (1.4)
Carcinosarcoma		4 (1.8)
Grade		
Low		127 (58)
High		92 (42)
Depth of invasion		
<50%		154 (70.3)
≥50%		65 (29.7)
Lymphovascular space invasion (LVSI)		
Absent		72 (32.9)
Focal		68 (31.1)
Substantial		79 (36.1)
Extensive necrosis *		
Absent		105 (47.9)
Present		113 (51.6)
Unknown/Not tested		1 (0.5)
MELF *		
Absent		147 (67.1)
Present		71 (32.4)
Unknown/Not tested		1 (0.5)
Tumor budding *		
Absent		128 (58.4)
Present		91 (41.6)
Lymph node status		
Negative		180 (82.2)
Positive		33 (15.1)
Unknown/Not tested		6 (2.7)
FIGO stage		
IA		124 (56.6)
IB/II		44 (20.1)
III		42 (19.2)
IV		9 (4.1)
Molecular subgroups		
*POLE*		17 (7.8)
MMRd		68 (31.0)
NSMP		88 (40.2)
p53abn		46 (21.0)
ESGO/ESTRO/ESP risk group		
Low		91 (41.6)
Intermediate		19 (8.7)
High–intermediate		23 (10.5)
High		77 (35.2)
Advanced/metastatic		9 (4.1)
Surgical approach		
Minimally-invasive	168 (76.7)	
Laparotomy	51 (23.3)	
Disease recurrence		
Absent		181 (82.6)
Present		38 (17.4)

* = 218 cases.

**Table 2 jpm-13-00723-t002:** Clinicopathologic characteristics of EC molecular subgroups. Values are counts (percentages) or mean ± standard deviation.

Characteristics	*POLE*	MMRd	p53abn	NSMP	*p*-Value
	(*n* = 17; 7.8%)	(*n* = 68; 31.0%)	(*n* = 46; 21.0%)	(*n* = 88; 40.2%)	
Age, years	57.8 ± 11.8	63.0 ± 9.4	66.8 ± 10.1	60.7 ± 10.3	0.002
Body mass index, kg/m^2^	26.4 ± 8.4	28.0 ± 7.1	26.0 ± 4.8	29.6 ± 7.9	0.036
Histotype					<0.001
Endometrioid	14 (82.4)	54 (79.4)	17 (37.0)	82 (93.2)	
Dedifferentiated/Undifferentiated	3 (17.6)	14 (20.6)	2 (4.3)	6 (6.8)	
Serous	0 (0.0)	0 (0.0)	20 (43.5)	0 (0.0)	
Clear cell	0 (0.0)	0 (0.0)	3 (6.5)	0 (0.0)	
Carcinosarcoma	0 (0.0)	0 (0.0)	4 (8.7)	0 (0.0)	
Grade					<0.001
Low	8 (47.1)	43 (63.2)	1 (2.2)	75 (85.2)	
High	9 (52.9)	25 (36.8)	45 (97.8)	13 (14.8)	
Depth of invasion ≥50%	3 (17.6)	24 (35.3)	19 (41.3)	19 (21.6)	0.047
LVSI					<0.001
Absent	5 (29.4)	15 (22.1)	11 (23.9)	41 (46.6)	
Focal	7 (41.2)	26 (38.2)	6 (13.0)	29 (33.0)	
Substantial	5 (29.4)	27 (39.7)	29 (63.1)	18 (20.4)	
Lymph node status					0.002
Negative	16 (94.1)	54 (81.8)	31 (68.9)	79 (92.9)	
Positive	1 (5.9)	12 (18.2)	14 (31.1)	6 (7.1)	
FIGO stage					<0.001
I	10 (58.8)	36 (53.0)	15 (32.6)	63 (71.6)	
IB/II	5 (29.4)	16 (23.5)	7 (15.2)	16 (18.2)	
III	2 (11.8)	13 (19.1)	19 (41.3)	8 (9.1)	
IV	0 (0.0)	3 (4.4)	5 (10.9)	1 (1.1)	
Extensive tumor necrosis					<0.001
Absent	4 (23.5)	27 (39.7)	17 (37.8)	57 (64.8)	
Present	13 (76.5)	41 (60.3)	28 (62.2)	31 (35.2)	
MELF					<0.001
Absent	11 (64.7)	33 (48.5)	40 (88.9)	63 (71.6)	
Present	6 (35.3)	35 (51.5)	5 (11.1)	25 (28.4)	
Tumor budding					0.017
Absent	9 (52.9)	31 (45.6)	25 (55.6)	62 (70.5)	
Present	8 (47.1)	37 (54.4)	20 (44.4)	26 (29.5)	
Mitoses/10 HPF	76.6 ± 35.9	55.5 ± 24.4	86.8 ± 43.9	32.2 ± 26.2	<0.001

**Table 3 jpm-13-00723-t003:** Multivariate Cox regression analysis of predictors of recurrence in NSMP subgroup.

Characteristics	Coefficients	Lower 95% CI	Upper 95% CI	Std. Error	z	*p*-Value
Endometrioid histotype	−2.73	−9.98	1.52	2.17	1.26	0.208
Dedifferentiated histotype	−4.31	−8.17	−0.46	1.97	2.19	0.028
Stage I	−2.51	−4.78	−0.23	1.16	2.16	0.031
Stage IB/II	−2.5	−4.69	−0.31	1.12	2.24	0.025
High grade	0.04	−3.21	3.29	1.66	0.02	0.982
Mitoses/10 HPF	0.00	−0.03	0.03	0.02	0.09	0.927
Extensive tumor necrosis	2.71	0.22	5.21	1.27	2.13	0.033
Substantial LVSI	−2.05	−4.14	0.03	1.06	1.93	0.054

CI: Confidence Interval; Std. Error: Standard Error.

**Table 4 jpm-13-00723-t004:** Multivariate Cox regression analysis of predictors of recurrence in early-stage NSMP subgroup.

Characteristics	Coefficients	Lower 95% CI	Upper 95% CI	Std. Error	z	*p*-Value
Endometrioid histotype	−15.53	−385.57	354.51	188.80	0.08	0.93
Dedifferentiated histotype	−2.66	−6.54	1.22	1.98	1.34	0.18
High grade	11.98	−358.05	382.01	188.79	0.06	0.95
Mitoses/10 HPF	0.01	−0.04	0.06	0.03	0.33	0.74
Extensive tumor necrosis	1.28	−1.23	3.78	1.28	1.00	0.32
Substantial LVSI	−2.99	−5.33	−0.64	1.20	2.50	0.01

CI: Confidence Interval; Std. Error: Standard Error.

## Data Availability

The datasets presented in this study can be found in online repositories. The names of the repository/repositories and accession number(s) can be found below: NCBI Bioproject, PRJNA932605.

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
