# Peer review of "Prognostic Impact of Pathologic Features in Molecular Subgroups of Endometrial Carcinoma"

_jpm, 2023, doi:10.3390/jpm13050723_

Round 1

Reviewer 1 Report

This study describes the prognostic role of clinicopathologic and molecular characteristics of EC. It is a well written manuscript. I have the following comments:

1.       In the methods section please add the number of patients with EC treated in the authors’ institution each year.

2.       Is this a prospective study or a retrospective analysis of prospectively collected data?

3.       When did the recruitment take place?

4.       How were the patients treated? Did all patients undergo surgery? There are patients with stage III and IV included in the study? How were these patients operated? I would suggest the separate table with the information regarding the treatment strategies.

Author Response

We thank the reviewers for their suggestions. Below are the point-by-point responses.

Reviewer 1:

This study describes the prognostic role of clinicopathologic and molecular characteristics of EC. It is a well written manuscript. I have the following comments:

  1. In the methods section please add the number of patients with EC treated in the authors’ institution each year.

Response: We thank the reviewer for the comments. We have added what was requested in the materials and methods section

  1. Is this a prospective study or a retrospective analysis of prospectively collected data?

Response: We thank the reviewer for the comments. We have added what was requested in the materials and methods section

  1. When did the recruitment take place?

Response: We thank the reviewer for the comments. We have added what was requested in the materials and methods section

  1. How were the patients treated? Did all patients undergo surgery? There are patients with stage III and IV included in the study? How were these patients operated? I would suggest the separate table with the information regarding the treatment strategies.

We thank the reviewer for the comments. We have added what was requested in the materials and methods section and also in table 1.

Reviewer 2 Report

This article shows the usefulness of the molecular classification in endometrial cancer. Authors also present the relevance of clinicopathological parameters with subclasses of molecular classification. Their study is not novel, but provides good information to clinicians. Minor points should be revised.

1.        Figure 1 will become better, if subgroup names of “POLE”, “MMRd”, “p53abn”, and “NSMP” would be added in the figure.

2.       Figure 4-8 should include p value in the figure legends.

Author Response

We thank the reviewers for their suggestions. Below are the point-by-point responses.

This article shows the usefulness of the molecular classification in endometrial cancer. Authors also present the relevance of clinicopathological parameters with subclasses of molecular classification. Their study is not novel, but provides good information to clinicians. Minor points should be revised.

  1. Figure 1 will become better, if subgroup names of “POLE”, “MMRd”, “p53abn”, and “NSMP” would be added in the figure.

Response: We thank the reviewer for the comments. We have added what was requested in the figure.

  1. Figure 4-8 should include p value in the figure legends.

Response: We thank the reviewer for the comments. We have added what was requested in the figure legends.